# Multianalyte serology in home-sampled blood enables an unbiased assessment of the immune response against SARS-CoV-2

Niclas Roxhed [1,2✉], Annika Bendes [3,11], Matilda Dale [3,11], Cecilia Mattsson[3,11], Leo Hanke [4], Tea Dodig-Crnković [3], Murray Christian[4], Birthe Meineke[5,6], Simon Elsässer [5,6], Juni Andréll[7], Sebastian Havervall[8], Charlotte Thålin [8], Carina Eklund[9], Joakim Dillner [9], Olof Beck[10], Cecilia E. Thomas [3], Gerald McInerney [4], Mun-Gwan Hong [3], Ben Murrell [4], Claudia Fredolini [3] & Jochen M. Schwenk [3✉]

Serological testing is essential to curb the consequences of the COVID-19 pandemic. However, most assays are still limited to single analytes and samples collected within healthcare. Thus, we establish a multianalyte and multiplexed approach to reliably profile IgG and IgM levels against several versions of SARS-CoV-2 proteins (S, RBD, N) in home-sampled dried blood spots (DBS). We analyse DBS collected during spring of 2020 from 878 random and undiagnosed individuals from the population in Stockholm, Sweden, and use classification approaches to estimate an accumulated seroprevalence of 12.5% (95% CI: 10.3%–14.7%). This includes 5.4% of the samples being IgG+IgM+ against several SARS-CoV-2 proteins, as well as 2.1% being IgG−IgM+ and 5.0% being IgG+IgM− for the virus' S protein. Subjects classified as IgG+ for several SARS-CoV-2 proteins report influenza-like symptoms more frequently than those being IgG+ for only the S protein (OR = 6.1; $p < 0.001$). Among all seropositive cases, 30% are asymptomatic. Our strategy enables an accurate individual-level and multiplexed assessment of antibodies in home-sampled blood, assisting our understanding about the undiagnosed seroprevalence and diversity of the immune response against the coronavirus.

[1] Micro and Nanosystems, School of Electrical Engineering and Computer Science, KTH Royal Institute of Technology, Stockholm, Sweden. [2] MedTechLabs, BioClinicum, Karolinska University Hospital, Solna, Sweden. [3] Science for Life Laboratory, School of Engineering Sciences in Chemistry, Biotechnology and Health, KTH Royal Institute of Technology, Solna, Sweden. [4] Department of Microbiology, Tumor and Cell Biology, Karolinska Institutet, Solna, Sweden. [5] Science for Life Laboratory, Karolinska Institutet, Department of Medical Biochemistry and Biophysics, Division of Genome Biology, Solna, Sweden. [6] Ming Wai Lau Centre for Reparative Medicine, Stockholm node, Karolinska Institutet, Solna, Sweden. [7] Science for Life Laboratory, Department of Biochemistry and Biophysics, Stockholm University, Solna, Sweden. [8] Division of Internal Medicine, Department of Clinical Sciences, Karolinska Institutet, Danderyd Hospital, Danderyd, Sweden. [9] Karolinska University Laboratory, Karolinska University Hospital, Stockholm, Sweden. [10] Department of Clinical Neuroscience, Karolinska Institutet, Stockholm, Sweden. [11]These authors contributed equally: Annika Bendes, Matilda Dale, Cecilia Mattsson. ✉email: roxhed@kth.se; jochen.schwenk@scilifelab.se

The infection with SARS-CoV-2 has been declared a pandemic by the World Health Organization (WHO), and diagnostics has become vital in fighting the disease[1]. In contrast to reverse transcription PCR (RT-PCR) detecting the viral RNA, serological assays reveal how the humoral immune system has coped with a previous infection. The first generation of enzyme-linked immunosorbent assay (ELISA) tests determined the antibody responses against the nucleocapsid protein (N), but tests with the trimeric spike glycoprotein (S) and its receptor-binding domain (RBD) have also become available[2–4]. Serological testing has given estimates for seroprevalence rates in several populations and has often included specific sub-populations[5–7], and assays testing the different SARS-CoV-2 proteins in parallel have recently emerged for a variety of platforms to capture the humoral immune response[8–12]. However, medical laboratory serological tests require venous blood drawn by healthcare professionals, and rapid self-tests using capillary blood currently do not fulfill the precision requirements[13,14]. Indeed, an attractive strategy would combine home- or self-sampling with precise laboratory analysis[15], especially when measures need to be taken to limit exposure and infections. Dried blood spots (DBS) is a well-established method to screen neonates for in-borne disease and has also been used to detect antibodies against viruses[16]. However, measurements of DBS are often not accurate enough[17], and self-sampling by patients is difficult leading to significant failure rates[18]. To address these issues, devices that simplify sampling and provide precise volumes have been developed[19,20].

Here, we distributed precision home-sampling kits to a random selection of households in Stockholm during the first wave of the COVID-19 pandemic. We analyzed the humoral immune response to SARS-CoV-2 infections in DBS by multiplexed serology assays (Fig. 1a) to demonstrate the precision and utility of self-sampling for unbiased but reliable estimations of the seroprevalence.

## Results

To achieve a reliable determination of the immune response against SARS-CoV-2 outside a clinical care setting, we established an analytical pipeline using DBS in combination with multi-analyte assays. After validating the novel approach against venous plasma, commercial ELISA assays, multiple constructs to represent several SARS-CoV-2 proteins were used to build the assay. DBS samples collected from households of the Stockholm population were then investigated for seropositivity with IgG and IgM levels, and seroprevalence frequencies were related with demographic information.

**A multiplexed COVID-19 serology assay with DBS.** As outlined in detail in the Supplementary Note 1 and depicted in Fig. S1, (i) qDBS blood collection cards were used to obtain exactly 10 μl of blood by finger pricking, (ii) the S, RBD, and N proteins were used as these appeared to be most immunogenic[12], and (iii) suspension bead array (SBA) assays were built for the multiplexed detection of IgG and IgM levels. As described in Supplementary Note 2, the carry-over from a serial release of the DBS samples was found to be acceptable (<15%; Fig. S2). The workflow had average intra-day CV of 13% (10–15%) and the inter-day CV 18% (12–22%) across all antigens (see Table S1 and Supplementary Note 3). The antibody detectability was checked and found to be within the analytical range (Fig. S3 and Supplementary Note 4). A longitudinal DBS analysis with a single donor found stable IgG levels against multiple antigens for at least 30 days after onset of symptom, while levels of IgM declined (Fig. S4). There was a high concordance in data using DBS and EDTA plasma, commercial ELISAs (S1 and N proteins), and different constructs produced

for the virus proteins (Figs. S5–S8). The SBA method was described for plasma elsewhere[21] and report levels of sensitivity and specificity >98%. This demonstrated the precision of the developed workflow and offered a reliable approach to determine seroprevalence levels of home sampled blood.

**Collecting self-sampled blood from random individuals of the Stockholm population.** To demonstrate the utility of our approach, two sets of 1000 blood sampling kits with questionnaires were distributed early and late April of 2020 by cold mailings to a blinded selection of inhabitants (age 20–74) in the urban area of Stockholm, Sweden. In total, 55% of the sampling cards were received back within 3 weeks, 44% ($N = 878$) were approved for analyses, hence 82% of the participating individuals succeeded with self-sampling. Sampling dates were inferred from the signed consent forms to range from mid-April to early May (Fig. 1b) and age ranges as well as sex matched with the population registers (Fig. 1c). As shown in Table S3, there were a slight difference in both compliance and sampling success between sexes, being higher for females. Questionnaire data showed that 13% self-reported fever under influenza-like symptoms, 22% listed symptoms related to issues with breathing, and >50% reported no symptoms. Nobody reported being tested positive for SARS-CoV-2 by PCR.

**Antigen-centric assessment of seroprevalence in home-sampled blood.** Levels of IgG and IgM antibodies against SARS-CoV-2's S, RBD, and N proteins as well as other relevant antigens (Tables 1 and 2) were determined in the 878 population DBS samples approved for analysis. The relative antibody levels obtained after processing the raw data are exemplified in Fig. 1d and panels of Fig. S9A–D. As expected, a highly skewed distribution of antibody levels were found for proteins from the SARS-CoV-2, and only a subset of all participants was seropositive for the new virus. For endemic viruses, such as Epstein–Barr virus (EBN_01) and hCoV-NL63 (NLS_01), a majority of the participants was expected to carry antibodies against these viruses, hence outlying samples indicated those that were presently seronegative.

To determine the SARS-CoV-2 seroprevalence, we applied a cut-off set at >6 times the standard deviation (SD) in relation to the most frequent antibody level determined among probable negative controls. Our model assumed that a majority of participants did not have antibodies against the virus. For each of the constructs representing the S, RBD, and N proteins, sensitivity levels of 100% were determined in 40 positive controls and specificity levels of 96–100% were determined in a set of 95 negative controls, see Supplementary Data 1 (sheet: "Prevalence per Antigen"). In the population study, the frequency of IgG-positive samples (denoted IgG+) ranged from 5.1 to 11.5% (Table S4), with the S proteins revealing more IgG+ compared to RBD or N proteins. There were 0–9.2% IgM-positive samples (denoted IgM+) with similar ranking concerning the SARS-CoV-2 proteins. For S1 proteins representing other coronaviruses (MERS-CoV, SARS-CoV-1, hCoV-NL63), positive and negative control samples remained negative for IgG or IgM. In the population samples, these S1 proteins showed a neglectable correlation with IgG levels of the SARS-CoV-2 proteins (rho < 0.3), and minor correlations with IgM levels (rho < 0.6) were driven by common trends in low antibody titers.

**Multianalyte-based assessment of SARS-CoV-2 seroprevalence.** To make the best use of the multianalyte data available for both antibody isotypes, we chose to perform unsupervised clustering (UMAP and PCA) and combinatorial analysis (dual antigen) to determine the seroprevalence. As shown in Fig. 2a, b for UMAP

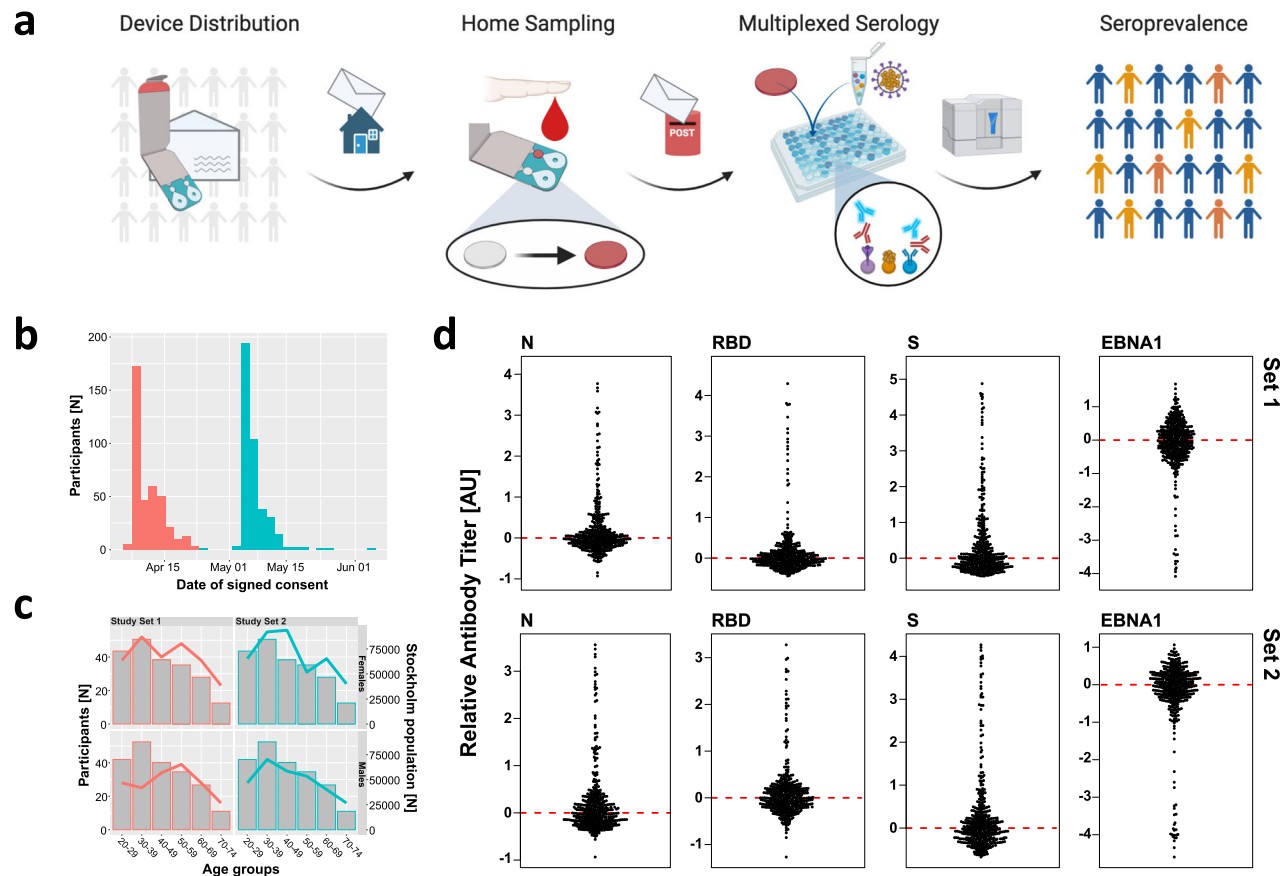

**Fig. 1 Translational approach for multiplexed serology in home-sampled dried blood spots. a** Blood collection devices were distributed by postal mail to collect blood samples from finger pricking at home. Cards were then mailed to the laboratory for multiplexed serological analysis. Data from antibody titers against multiple virus proteins was used to stratify individuals for seroprevalence. **b** The histogram shows the number of participants (y-axis) donating blood for the study set 1 (red bars) and set 2 (petrol bars) with inferred date of sampling (x-axis). **c** Comparison of sex and age-range demographics between our two study sets (lines) and the Stockholm population statistics (bars). **d** A panel of the relative IgG antibody levels detected in the two study sets against the SARS-CoV-2 proteins S, RBD, and N as well as EBNA1. The dashed red lines indicate the center of the normalized assay data. Source data are provided as a Source Data file.

**Table 1 Proteins from SARS-CoV-2.**

| Protein | Acronym | Provider | Product ID/Note | Lot/Batch |
|---|---|---|---|---|
| S | SPK_01 | McInerney lab | Foldon, His-tag, Strep-tag II | Batch 1 |
| S | SPK_02 | McInerney lab | Foldon, His-tag, Strep-tag II | Batch 2 |
| S | SPK_03 | Andréll lab | StrepIIHis-tag | Batch 1 |
| S | SPK_04 | Andréll lab | His-tag | Batch 1 |
| RBD | RBD_01 | McInerney lab | FLAG-tag | Batch 1 |
| RBD | RBD_02 | McInerney lab | FLAG-tag | Batch 2 |
| RBD | RBD_03 | McInerney lab | His-tag | Batch 1 |
| RBD | RBD_04 | Andréll lab | His-tag | Batch 1 |
| N | NCP_01 | SinoBiological | 40588-V08B | LC14MC0309 |
| N | NCP_02 | Elsässer lab | 2xStrep-tag | Batch 1 |
| N | NCP_03 | Acro biosystems | NUN-C5227 | S34-2048F1-RB |
| S | SPS_01 | SinoBiological | 40591-V08H | LC14MC1012 |

**Table 2 Proteins from other viruses.**

| Protein | Acronym | Provider | Product ID/Note | Lot/Batch |
|---|---|---|---|---|
| SARS-CoV-1 S1 | SRS_01 | SinoBiological | 40150-V08B1 | LC14AP1505 |
| MERS-CoV S1 | MRS_01 | SinoBiological | 40069-V08B1 | LC12AU0205 |
| hCoV-NL63 S1 | NLS_01 | SinoBiological | 50600-V08H | LC14AP2005 |
| EBNA1 | EBN_01 | Abcam | ab138345 | GR3235466-1 |

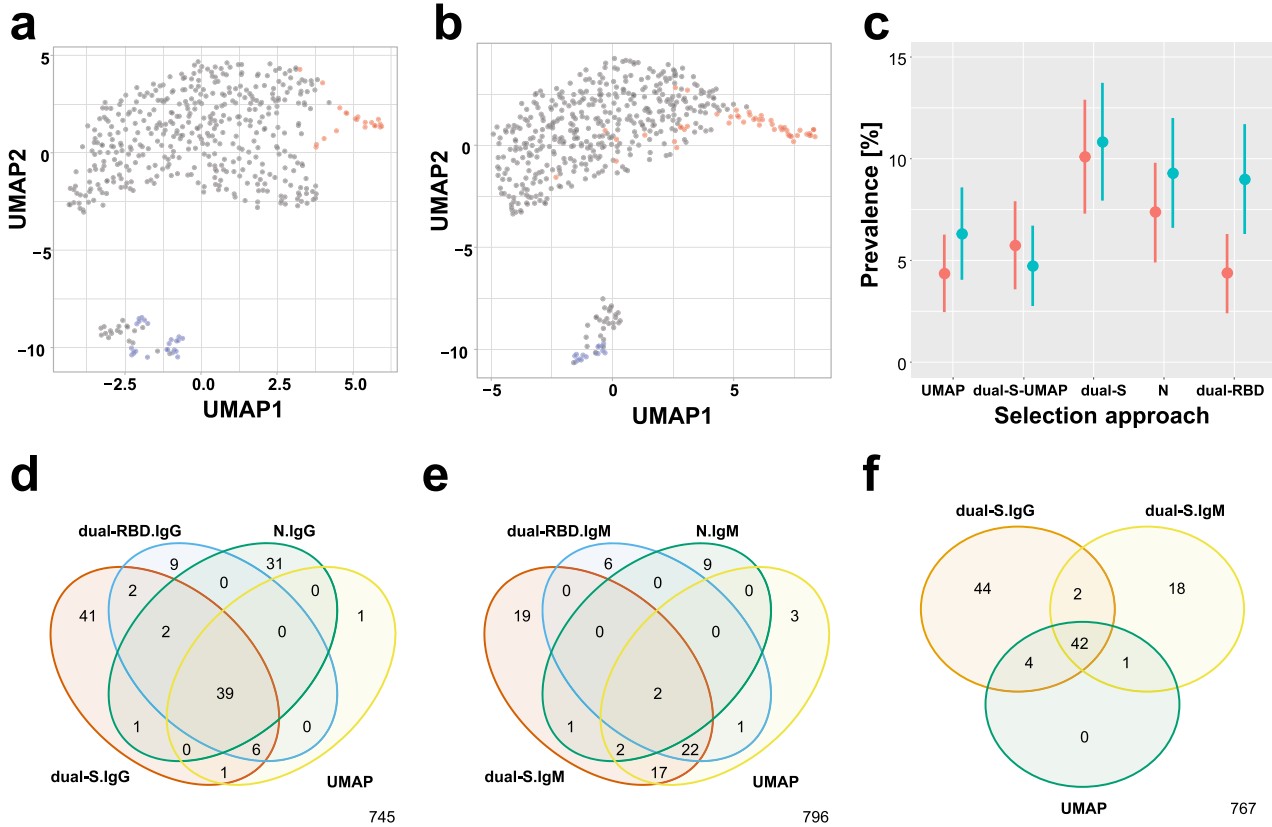

**Fig. 2 Seroprevalence and diversity of immune response. a, b** The UMAP (Uniform Manifold Approximation and Projection) clustering method was used to identify samples in study set 1 (**a**) and set 2 (**b**) with antibodies against SARS-CoV-2, data from IgG and IgM, as well as the included S, RBD, and N proteins were combined. The color codes indicate the samples from the population (gray), negative controls collected prior to 2020 (red), and PCR-confirmed positive controls (blue). UMAP parameters for **a** were n_neighbors: 25, min_dist: 0.5, seed: 42 and for **b** were n_neighbors: 27, min_dist: 0.5, seed: 254. **c** Seroprevalence determine by different selection methods for study set 1 (red) set 2 (petrol): UMAP clustering, overlap in IgG response against two S proteins (dual-S), excluding the UMAP from dual-S group (DUAL-S-UMAP), N protein, and overlap in IgG response against two RBD proteins (dual-RBD). The data are presented as percent seroprevalence with 95% confidence intervals. **d–f** Venn diagrams presenting the overlap in samples classified by the four selection methods as seropositive for **d** IgG and **e** IgM, as well as **f** comparing the samples deemed positive for IgG and IgM using the dual-S approach. Source data are provided as a Source Data file.

clustering (see Fig. S10A, B for PCA), two distinct clusters emerged in both study sets resembling a pattern found in the pilot study used for method validation (Fig. S8). The smaller clusters contained all convalescent PCR-positive donors, no pre-pandemic, as well as 19 samples from set 1 (4.4%; 95% CI: 2.5–6.3%) and 28 samples from set 2 (6.3%; 95% CI: 4.1–8.6%). The remaining samples clustered with the pre-pandemic controls. Antibody levels detected in these 47 UMAP-seropositive participants were elevated for multiple of the SARS-CoV-2 proteins (Fig. S11A, B). During the period mid-April and May of 2020, we found that 5.4% (95% CI: 4.1–8.6%) in our study population had similar antibody levels as convalescent, symptomatic PCR-confirmed cases. This frequency was slightly below the 7.4–10.2% reported by the Public Health Agency of Sweden[22] who tested patients in the primary care in Stockholm from April 27 to May 24 with the SBA method.

To enable a further increase in stringency in calling seropositivity for individual antigens, we used a combinatorial approach that required antibody levels for at least two recombinant versions of the S, RBD, or N proteins to be above cut-off (denoted dual-antigen). The frequency of dual-S IgG+ samples increased from 10.1% (95% CI: 7.3–12.9%) to 10.8% (95% CI: 7.9–13.7%), and for dual-RBD IgG+ from 4.4% (95% CI: 2.4–6.3%) to 9.0% (95% CI: 6.3–11.7%) between study set 1 and 2 (Fig. 2c). Only one N protein was common in the two study sets

and 7.4% (95% CI: 4.9–9.8%) of the samples in set 1 and 9.3% (95% CI: 6.6–12.0%) of the samples in set 2 were deemed as IgG+. The latter frequency decreased to 8.0% when using the dual-N approach in study set 2. In essence, using two representative antigens per SARS-CoV-2 protein slightly reduced seroprevalence levels by ~1%, but it supports an increased reliability.

**Antibody isotypes as indicators of time since infection.** Next, we investigated immune response per antibody isotype and used the data to judge the time elapsed since infection and onset of symptoms. We chose IgG−IgM+ to represent the early and IgG+IgM− to the later phases of the infection. As shown in Fig. 2d, e and Fig. S12A, more samples were classified as seropositive by IgG than IgM, and as discussed above, most samples were positive for the S protein. Six samples were IgG−IgM+ by the dual-RBD and nine were IgG+IgM−. For the N protein, there were another nine IgG−IgM+ samples and 31 IgG+IgM− samples. The overlap of individual preparations of SARS-CoV-2 proteins is shown in Fig S12B–E, and this illustrated some of the facets of immune responses. Figure 2f compares the samples assigned seropositive by the dual-S approach for IgG, IgM, and UMAP. On top of the 46 UMAP-seropositive samples, there were 2.1% (18/878) IgG−IgM+ samples and 5.0% (44/878) IgG+IgM− samples, as well as two additional IgG+IgM+ samples. Cumulatively, there were 110 samples deemed seropositive, thus representing 12.5%

(95% CI: 10.3–14.7%) of the population. This number is well in line with the range of 7.4–10.2% reported by the Public Health Agency of Sweden[22], and likely deviating due to an additional 2% found while analyzing IgM levels.

**Relationship between multianalyte seroprevalence and reported symptoms**. Lastly, we investigated the relationship between seroprevalence and demographics. In the UMAP-seropositive group ($N = 47$), significantly more participants ($p < 0.01$) reported fever or severe influenza-like symptoms (OR = 6.7; $p < 0.001$) and symptoms related to breathing (OR = 2.4; $p = 0.007$) (Table S5). This group was younger ($p = 0.006$) and only 17% (8/47) reported no influenza-like symptoms, while 60% (28/47) reported no symptoms related to breathing. In contrast, 60% (505/831) of the seronegative group reported no influenza-like symptoms and 79% (657/831) reported no symptoms related to breathing. There were similar but less significant differences for the dual-S group ($N = 92$) in terms of age ($p = 0.1$), influenza-like symptoms (OR = 2.5, $p < 0.001$), and breathing (OR = 1.8; $p = 0.01$) (Table S6). Excluding 45 UMAP-seropositive subjects in common with the dual-S group left a set of 46 (5.2%; 95% CI: 3.8–6.7%) population participants being IgG$^+$. These 46 were slightly older than the UMAP-seropositive group ($p = 0.03$) and reported either no or only mild influenza-like symptoms (OR = 6.1; $p < 0.001$) (Table S7). This investigation pointed at a relationship between severity of symptoms and the number of co-detected SARS-CoV-2 proteins. Notably, 30% (33/110) of all seropositive subjects remained asymptomatic.

## Discussion

We conducted an unbiased translational serology approach using blood from finger prick samples. The developed procedure was on par with data from venous blood samples and commercial ELISA assays, and it allowed to accurately determine the antibody prevalence against several SARS-CoV-2 proteins in the Stockholm population during the first wave of the COVID19 pandemic. The use of several constructs in multiplexed assays with DBS delivered seroprevalence levels that matched the ELISA-based estimates determined from clinically collected samples[23]. From the available questionnaire data, a link between disease severity and the number of detected antigens could be drawn alongside the observation that 30% of the seropositive participants reported no symptoms.

The use of dried blood is a well-established procedure that offers many logistical benefits. During the 2020 COVID-19 pandemic, sampling by laypeople outside the hospital and healthcare setting was considered to be particularly attractive because it reduced the burden of healthcare centers to test patients for SARS-CoV-2 infections. Home-sampling also reduced the need for participants or patients of risk groups to leave their homes. In addition, inviting participants via cold mailings reduces the bias compared to studies where participation is self-enrolled[24]. However, home-sampling poses challenges concerning the lack of compliance, experience with self-sampling, and quality of the blood sample. We observed a high 82% success rate of self-sampling and incomplete blood collection was found only occasionally.

The multiplexing technology enabled us to determine the presence and levels of antibody against several different virus analytes in each DBS sample under identical analysis conditions. It also allowed to account for individual-specific background levels of antibodies binding to the beads and measure a large number of samples even if only limited amounts were available. We further used different preparations of SARS-CoV-2 proteins, epitope tags, protein coupling chemistries, and production batches to increase the validity of our observations by utilizing the overlap between serological profiles against several proteins. Multiplexing also allowed the inclusion of antigens from endemic or new human coronaviruses, such as hCoV-NL63, MERS-CoV, and SARS-CoV-1, or other the prevalent viruses such as Epstein–Barr virus. The observed antibody reactivities towards SARS-CoV-2 were discrete and not reflected by cross-reactivity for these virus proteins. As also shown by two independent studies using plasma and SBA assays[21,25], misclassification and cross-reactivity due to the tested viruses was minimal.

We detected IgG and IgM antibodies against multiple proteins in all of the convalescent positive controls as well as 12.5% of the tested population. Since this study was conducted during the very early phase of the pandemic, when analytical tools and standards still remained limited, we chose to use concordance between the different virus proteins to increase the certainty about the seropositivity. The self-reported information about health conditions during the previous months included questions concerning influenza-like symptoms (fever, coughing, and breathing difficulties). No questions related to loss of taste or smell had been included in our questionnaire even though these symptoms commonly emerged[26]. It was not possible to request additional information from the anonymous donors. Interestingly, there was a correlation between the severity of the self-reported symptoms (influenza and breathing) and the number of recognized SARS-CoV-2 proteins. This could point at a stronger current immune response against the infection. With an additional 2% only positive for IgM and 5% only positive for IgG, we identified donors at early and late stages of the infection. This meant that among the 110 subjects deemed seropositive of the 878 tested persons, 16% were recently infected, 44% about 14 days post infection, and another 40% already 30 days post infection.

We found discrepancies between the humoral immune response against the three most commonly used SARS-CoV-2 proteins. As recently shown, the analysis of several antibody isotypes against different proteins does indeed provide a more complete picture about COVID-19 infections[27]. The presented SBA assays were built on the SARS-CoV-2 proteins most commonly used in serological analyses, with a focus on the main neutralizing antibody response mounted against the viral S protein, and especially against its RBD. While antibodies against RBD were almost always accompanied by antibodies against S, not that many with antibodies against the S protein also had antibodies against RBD. Accordingly, the S protein was found to reveal the highest seropositivity rates, which was mostly complementarity for both IgG and IgM detection. Reactivity towards RBD was common for both antibody isotypes, while only IgM was in some donors. Interestingly, IgM was found only in a few subjects when using the N proteins.

Presenting a variety of approaches to determine the seroprevalence provided different perspectives to the time since infection as well as the emerging relationship between the severity of symptoms and the number of antigens recognized by the humoral immune response. The observed diversity of serological profiles may provide additional pointers that ask for a wider assessment of humoral response for SARS-CoV-2, thus expand our insights about the infection. Data from longitudinal serology studies are now starting to emerge[28]. These indicate that antibody against the virus remain elevated even months after the infection, and that the S protein presents a more sensitive antigen than the N protein[29]. In addition, other studies revealed the importance of the S protein and its RBD domain in the context of neutralization of the virus[30]. This adds to the notion that multiplexed data will provide a valuable component to assist our understanding of different forms of COVID-19, which can also be captured by detecting the levels of IgG, IgM, or IgA.

The study design was to mail out blood collection kits to random individuals in Stockholm who represent the city's demographics in term of age and sex, but without considering health, travel, or socioeconomic factors. Stockholm was the most COVID-19-affected region in Sweden during the first wave of the pandemic, and both PCR testing capacities and confirmed cases were increasing. According the Public Health Agency of Sweden, there were ~4500 PCR-confirmed cases as per April 12 and ~10,000 as per May 10 (ref. [31]). Extrapolating our serological data to the 1.25 million people living in the included areas of Stockholm estimates that ~150,000 inhabitants would have been infected with SARS-CoV-2 during our study period. However, we refrain from drawing further conclusions from these population estimates because PCR testing remained limited and focused on those with symptoms, while our study was comparably underpowered.

The combination of precise home sampling of blood and laboratory-based serological analysis can become a viable approach to assess the humoral immune response. This strategy can be applied for large cohort studies with well-informed participants[32], and as demonstrated here, also engage random donors from the general population. Expanding the capabilities of self-sampled blood beyond clinical serology, such as by integrating other advanced analytical pipelines, will open up the utility of home sampling concepts for further health tests in the time after the current pandemic.

## Methods

**Samples**. *Pilot study for method validation.* Venous as well as capillary blood samples were collected from volunteers ($N = 50$) among personnel at a healthcare center in Stockholm between May 14 and 18, 2020 by a trained phlebotomist (Table S2). Venous blood samples (two per donor) were collected through venipuncture into EDTA blood collection tubes (K2E K2EDTA Vacuette tube, #454410, Lot# A19104MX, Greiner Bio-One) and capillary blood samples were obtained by finger-pricking using a contact-activated lancet (BD Microtainer #366594, BD) and applying blood droplets onto a quantitative DBS sampling card (qDBS, Capitainer AB, Stockholm, Sweden) according to the supplier's instructions. After blood collection, one of the venous blood tubes was centrifuged and the blood plasma was collected into a separate tube. Both the plasma sample and the other blood tube was stored at −20 °C until further use. The qDBS cards were stored at room temperature until heat treatment prior to extracting the blood-filled discs.

*Population DBS study.* Capillary blood samples from the general population were obtained by cold-mailing home-sampling kits to 2000 randomly selected inhabitants (20–74 years old) from the population register in metropolitan Stockholm (Table S3) during April 2020. Stockholm municipalities included Upplands-Väsby, Järfälla, Ekerö, Huddinge, Botkyrka, Salem, Tyresö, Täby, Danderyd, Sollentuna, Stockholm, Nacka, Sundbyberg, Solna, and Lidingö with a total of 1,227,713 inhabitants aged 20–74 years. We examined each sampling card by visual inspection for the degree that the discs were filled with blood. Only the questionnaire data from donors providing at least one disc filled with 10 µl of blood were then analyzed.

Home-sampling kits were mailed in standard C4 envelopes containing the kit with a contact-activated lancet (BD Microtainer #366594, BD), quantitative DBS sampling card (qDBS, Capitainer AB, Stockholm, Sweden), return pouch (Capitainer AB, Sweden), alcohol swab, gauze and band-aid, as well as an information letter, questionnaire, consent form, C5 prepaid return envelope, and an instruction sheet for home-sampling (MM20-009-01, Capitainer AB, Sweden). Individuals who volunteered to participate in the study were asked to perform self-sampling according to the instructions and return the filled sampling card, questionnaire, and consent form in the prepaid return envelope by regular mail. The samples were analyzed within 3–4 weeks after receiving the last blood cards.

*Control DBS samples.* Besides the samples used in the pilot study for method validation we added control samples to the population study from donors with known exposure to the virus. As negative controls for set 1, we used 25 DBS samples from venous blood donors. Samples were collected from anonymous donors prior to 2020 and purchased from a blood bank (Blodcentralen, Region Stockholm). For set 2, we used 44 capillary blood DBS samples collected in a biobank before 2019 from patients (38–77 years old; 75% male) from the Stockholm region. In addition, we used commercially available EDTA plasma from a pool of anonymous healthy males and females (#HMPLEDTA2, Lot# BRH1176237, Seralab) that had been purchased prior to the COVID-19 outbreak and was stored at −80 °C until use. As positive controls we used EDTA plasma

samples from four random COVID-19-convalescent and PCR-confirmed individuals obtained from Karolinska University Hospital in Huddinge (no metadata available). All participants had recovered from PCR-verified COVID-19 since at least 2 weeks. Ten microliters of plasma were loaded directly onto each disc of the DBS cards. In addition, we obtained capillary blood samples from five COVID-19-convalescent and PCR-confirmed individuals (30–69 years old; 60% male; all reported influenza-like symptoms and reported breathing issues) who volunteered to donate blood after hospital discharge and using the home-sampling procedure as above. Further to this, one COVID-19-convalescent and PCR-confirmed individual volunteered to donate capillary blood samples every 3–8 days during a 3-week period (no metadata available). Samples from ELISA and PCR-positive participants from the pilot study were used as additional controls.

All blood donors gave informed documented consent. The study was approved by the regional ethical board (EPN Stockholm, Dnr 2015/867-31/1) and the Swedish Ethical Review Authority (EPM, Dnr 2020-01500). Use of biobanked controls samples was approved by the Swedish Ethical Review Authority (Dnr 2020-02483). At Karolinska University Hospital in Huddinge, corona serology testing as part of a convalescent plasma donation study was approved by the Swedish Ethical Review Authority (EPM, Dnr 2020-01479). All cards were barcoded and stored at room temperature until use, or as stated otherwise.

**Protein production**. Recombinant proteins were either obtained from commercial providers or produced by the independent labs as follows and as summarized in Table 1. The following acronym codes were used for the different proteins and batches: For the spike ectodomain we used SPK, for the receptor binding domain we used RBD, and for the nucleocapsid proteins we used NCP.

*S proteins.* The McInerney lab obtained the plasmid for the expression of the SARS-CoV-2 prefusion-stabilized spike ectodomain from Wrapp et al.[33], as a gift from Jason McLellan at University of Texas, USA. The plasmid encoding the SARS-CoV-2 spike ectodomain (GeneBank: MN908947) followed by T4 fibritin trimerization motif, a 8xHIS tag, and a StrepII-tag was used to transiently transfect FreeStyle 293F cells using FreeStyle MAX reagent (Thermo Fisher). The S1 ectodomain represented the residues 14–1208 (excluding the signal sequence and tags). The protein was purified from filtered supernatant on Streptactin XT resin (IBA Lifesciences), followed by size-exclusion chromatography on a superdex 200 in 5 mM Tris pH 8, 200 mM NaCl, and rebuffered into PBS before coupling to beads. The protein was produced on different dates as two batches following the same protocol, hence denoted SPK_01 and SPK_02.

The Andréll lab produced two spike ectodomain constructs (GeneBank: MN908947). The Sfoldon-His-StrepIIHis protein (SPK_03) is the same construct as SPK_01/02 mentioned above. A second spike trimeric ectodomain (SPK_04, Sfoldon-His) was generated using a plasmid provided as a kind gift from John Briggs and Andrew Carter at Laboratory of Molecular Biology MRC, UK, see ref. [34]. SPK_04 was therefore a modified version of SPK_01/02 and SPK_03 representing the residues 14-1211 (excluding the signal sequence and tags). Expi293 were transiently transfected with SPK_03 or SPK_04 using PEI transfection reagent (# 23966, Polysciences). After 72 h post-transfection, the supernatant was cleared and Spike ectodomain purified on Ni-NTA resin (#88221, Thermofisher). For SPK_04 protein the Ni-NTA step was followed by size-exclusion chromatography on a Superose 6 gel filtration column in 20 mM HEPES, 200 mM NaCl. For SPK_03 protein the Ni-NTA step was followed by purification on Strep-Tactin XT resin (#2-4010-010, IBA) prior to gel filtration on a Suprose 6 gel filtration column in 20 mM HEPES, 200 mM NaCl.

*RBD proteins.* The McInerney lab prepared the two RBD constructs termed RBD_01 and RBD_03 (GeneBank: MT380725.1). RBD domain RVQ-VNF of SARS-CoV-2 (RBD_01) was cloned upstream of an enterokinase cleavage site and a human FC. This plasmid was used to transiently transfect FreeStyle 293F cells using the FreeStyle MAX reagent and this FC fusion was purified from filtered supernatant on Protein G Sepharose (GE Healthcare). The protein was cleaved using bovine enterokinase (GenScript) leaving a FLAG-tag on the C-terminus of the RBD. Enzyme and FC-portion was removed on His-Pur Ni-NTA resin and Protein G sepharose, respectively, and the RBD was purified by size-exclusion chromatography on a Superdex 200 in 5 mM Tris pH 8, 200 mM NaCl. This protein was termed RBD_01 and a second batch, produced in the same way, but on a different day, was termed RBD_02. For the second variant, the RBD domain RVQ-QFG (RBD_03) was cloned upstream of a Sortase A recognition site and a 6x His-tag, and expressed in FreeStyle 293F cells as above. RBD_03 was purified from filtered supernatant on His-Pur Ni-NTA resin, followed by size-exclusion chromatography on a Superdex 200.

The Andréll lab prepared RBD_04 by using the RBD-His plasmid obtained from BEI resources NR52309 (GeneBank: MN908947)[4]. Expi293 cells were transiently transfected with RBD_04 using using PEI transfection reagent (# 23966, Polysciences). After 72 h post-transfection, the supernatant was cleared and RBD_04 purified on Ni-NTA resin (#88221, Thermofisher) followed by size-exclusion chromatography on a Superdex 200 gel filtration column in PBS.

*N proteins.* The Elsässer lab prepared the nucleocapsid protein denoted NCP_02. The mammalian expression plasmid pLVX-EF1alpha-nCoV2019-N-IRES-Puro used for mammalian expression plasmid was a kind gift from Nevan Krogan lab at UCSF. HEK293T cells were transfected using PEI. Cells were harvested 60 h post-transfection and NCP protein affinity purified similar as

described elsewhere[35]: cells were washed with PBS and lysed in 50 mM Tris, 150 mM NaCl, pH 7.4, 1 mM EDTA, 0.5% NP-40 supplemented with 1× complete protease inhibitor (#11873580001, Roche). Lysate was cleared by centrifugation and incubated with 20 μl 5% Strep-Tactin bead suspension (2-4090-002, IBA), 30 min on ice. The resin was washed 3× with 100 mM Tris/HCl, pH 8.0; 150 mM NaCl; 1 mM EDTA, 0.05% NP40 and eluted in the same buffer supplemented with 50 mM biotin.

**Experimental study design.** *Pilot study for method validation.* For the ELISA analysis, EDTA plasma (N = 50) and DBS eluates prepared from finger pricked sampling (N = 38) were analyzed together as described below. The multiplexed serology performed with the SBA assays were conducted in duplicate. The latter assay analyzed EDTA plasma (N = 50), DBS eluates of whole blood collected by finger pricking (cDBS, N = 50) as well as from DBS eluates of whole blood prepared by applying blood to the cards that had been collected from venous draw (vDBS, N = 50).

*Population study.* One disc from each card was transferred into 96-well plates in the order the cards were returned my mail. Each 96-well plate was filled with 80 discs.

For the first set of population samples (Study Set 1, N = 435), each 96-well assay plate had two empty filter-discs, two wells with assay buffer only, four discs from negative controls prepared from the pool of EDTA plasma collected prior 2019, four negative controls consisting of DBS collected before the outbreak, and four positive controls in form of EDTA plasma samples applied to discs from COVID-19-convalecent donors. One well per plate was left empty for plate identification and orientation. The four control wells with healthy plasma had the same (pooled) sample, which provided data about reproducibility.

For the second set of population samples (Study Set 2, N = 443), each 96-well assay plate had two wells with assay buffer only and two discs from two PCR-positive individuals that were present in all assay plates. In addition, the second discs of eight previously analyzed subjects from study set 1 and second discs of four individuals from the pilot study were added. We also included and reanalyzed ten DBS eluates prepared for Study Set 1 as well as ten DBS eluates prepared for this study set.

**Assays**

*Sample preparation.* First, the blood sampling cards were heated at 56 °C for 60 min in an oven (UN55m, Memmert GmbH) in sets of 50. Each card was visually inspected to determine if at least one paper disc was correctly filled with blood. From each card deemed successful, one paper disc was ejected using a semi-automated card-punching apparatus (qDBS Card Puncher, Capitainer AB, Sweden) into one well of a flat bottom 96-well plate (#734-2327, VWR). To reduce contamination between cards, the puncher's blade was cleaned with a H$_2$O$_{dd}$-wetted, and then a 70% EtOH-wetted, synthetic swab for every new row of the 96-well plate. The transferred discs were then subjected to 100 μl of PBST containing 1× PBS with 0.05% Tween20 and protease inhibitor cocktail (#04693116001, Roche). The discs were then incubated under gentle shaking (170 r.p.m.) for 60 min at room temperature. The plates were then centrifuged for 3 min at 2100 × g (Allegra X-12R, Beckman Coulter Inc.). A supernatant of 70 μl was transferred into a PCR plate (#732-4828, VWR) and sample eluates were stored at −20 °C after the analysis. Protein concentrations of the eluates were determined using a Nanodrop spectrophotometer system (ND-1000, ThermoFisher) by measuring the absorbance at 280 nm in 2 μl per samples. The eluates were measured in triplicates and using the elution buffer as blank.

*Multiplexed serology assays.* Proteins were covalently coupled to color-coded magnetic beads (MagPlex, Luminex Corp.) using either NHS/EDC coupling as described elsewhere[36] or an Activation Kit For Multiplex Microspheres (A-LMPAKMM-10RXN, SigmaAldrich) for proteins stored in Tris-based buffers. Anti-human IgG (309-005-082, Lot # 132463, Jackson ImmunoResearch), anti-human IgM (IGM, 109-005-129, Lot # 147777, Jackson ImmunoResearch), and anti-human IgA (IGG, GA-80A, Lot # 0017, Immune Systems Ltd) antibodies were diluted to 1.8 μg/ml and coupled using NHS/EDC chemistry. These antibodies were used as controls in the assays. The beads were then mixed to create a suspension antigen bead array. Conjugation was confirmed using epitope-tag specific antibodies.

DBS eluates in 96-well plates were transferred to 384-well plates for the serological analysis. Eluates were diluted 1:2.5 in assay buffer containing 1× PBS with 0.05% Tween20 with 3% BSA (B2000-500, Lot# 08C5415, Saveen Werner) and 5% milk powder (70166-500 G, Lot# BCBT8091, Sigma-Aldrich). Negative and positive control plasma samples were diluted in assay buffer 1:50 and 1:7.5, respectively. Per diluted sample, 35 μl were then incubated with 5 μl antigen bead array for 1 h at room temperature, shaking at 650 rpm, dark, followed by washing in 3 × 60 μl PBS-T 0.05% using an automated washing system (Biotek EL406). The beads were then resuspended in 50 μl detection buffer containing either anti-human IgG-R-PE (H10104, Lot# 2079224, Invitrogen) diluted to 0.4 μg/ml in PBS-T 0.05% or anti-human IgM-R-PE (#109-116-129, Lot# 137465, Jackson Immunoresearch) diluted to 1 μg/ml in PBS-T 0.05%. The beads were then incubated for 30 min at room temperature under rotational shaking at 650 r.p.m. in the dark. Prior to performing the readout, the plates were washed 3 × 60 μl with PBS-T 0.05%, and 60 μl PBS-T were added into each well. The data were reported as median fluorescence intensity (MFI) values per antigen and sample. For each of

the data points, at least 32 events per bead ID were collected. Data were collected on Luminex FlexMap 3D instruments (Luminex) operated by the xPONENT software version 4.3.

*ELISA.* Seropositivity levels for human IgG were determined for S1 protein (EI 2606-9601G, Lot# E200428BX, EuroImmun AG) and the N protein (EI 2606-9601-2G, Lot# E200429BO, EuroImmun AG) according to the kit provider. Plasma samples were diluted 1:101, and DBS eluates were diluted 1:5 in the provided assay buffer.

**Data analysis**

*Data processing.* Data processing were performed in v.3.6.0 of R[37] or Julia[38]. For the SBA data of the study sets 1 and 2, MFI values were log transformed and normalized to adjust for background binding of human IgG as follows: A trend line was fitted to the bulk data which was assumed to represent the more frequent seronegative samples by regressing each protein profile against those reported for the internal negative control. This control was one population of beads subjected to the coupling procedure without the addition of any proteins, denoted as bare beads. To minimize the influence of the less frequent seropositive samples, we used a robust regression model with a Huber loss function[39] consisting of an L2 loss for errors smaller than 0.1 and L1 loss for larger errors, as well as an L2 penalty on regression parameters. This procedure was implemented using the Julia package MLJLinearModels.jl and applied to normalize the data on the basis of the assay background obtained from the bare bead with the peak of the seronegative samples centered at zero. This normalization converted raw MFI data to residuals from this regression and denoted as relative antibody titers.

*Seroprevalence analysis.* The prevalence was estimated based on 6× SD added to the peak value of density plot applying Gaussian smoothing. The SD was the standard deviation of the values excluding 20% far from the density peak. For calculating the confidence interval (CI) of 95%, we assumed each set was a random sample from Stockholm population and applied normal approximation. Positive predictive value (PPV) was computed assuming the prevalence of the disease was 10%. When computing values for sensitivity, specificity, and PPV, repeated measurement of positive/negative control samples were not taken into consideration. The UpSet plots were created by "UpSetR v. 1.4.0" R package.

*Dimensionality reduction.* PCA and UMAP analysis were performed using the R packages "stats v. 3.6.0" and "umap v. 0.2.4.1" (https://CRAN.R-project.org/package=umap), respectively. For UMAP analysis a range of values for the hyperparameters "n_neighbors" and "min_dist" were tested with three repeats for each parameter combination. For each data set an UMAP layout representing the whole collection was chosen for visualization.

*Statistical analyses.* Data analyses were performed in v.3.6.0 of R[37]. Wilcoxon rank sum tests were used to determine the nominal p values for the variables age and influenza-like symptoms. Fisher's exact tests were used for the binary variables sex and breathing. Both tests were two-sided. Odd ratios (OR) were calculated for the seropositive groups of females, those reporting no influenza-like symptoms and no issues related to breathing.

**Reporting summary.** Further information on research design is available in the Nature Research Reporting Summary linked to this article.

## Data availability

Normalized and anonymized serology data can be made available for validation purposes and upon reasonable request to the corresponding authors. Source data are provided with this paper.

## Code availability

Analysis scripts are available at the GitHub repository for the Schwenk Lab[40]

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

## Acknowledgements

We thank all anonymous blood donors who volunteered to help us with this research project. We also thank A. Benckert, C. Stenfelt, A. Foley, T. Lundén, J. Lundén, E. Roxhed, V. Wennergren, and X. Tian who volunteered to pack envelopes; Drs. C. Linder, A. Pohanka, and S. Rosenborg from Karolinska Hospital and Dr. S. Rautiainen-Lagerström at Danderyds hospital for assisting in obtaining samples. At SciLifeLab, we thank Dr. A. Olsson, Dr. M. Michel, K. Mamonov and F. Ortis for their continuous efforts on generating proteins and peptides. We thank A. Martinez Casals for her kind assistance with processing the blood cards, and everyone at the Translational Plasma Profiling and Autoimmunity Profiling Facilities at SciLifeLab in Stockholm for their support. Everyone at the Human Protein Atlas Project is acknowledged for their tremendous efforts. Figure 1a and S1 were created with BioRender.com".This study was supported by grants to Science for Life Laboratory from the Knut and Alice Wallenberg Foundation (2020.0182) for "Translational Serology", funds from the Erling-Persson foundation for KTH Center for Precision Medicine (KCAP), KTH and Science for Life Laboratory. L.H., B. Murrell, and G.M. are supported by an EU grant (CoroNAb).

## Author contributions

A.B., M.D., and C.M. generated the immunoassay data and contributed to the experimental study design. L.H., B. Meineke, S.E., and J.A. produced and provided viral antigens. S.H., C.T., C.E., and J.D. provided clinical samples. M.C. and B. Murrell developed the normalization approach. T.D.-C., C.E.T., and M.-G.H. contributed to the experimental study design and analyzed the data. B. Murrell and G.M. provided expertise on virology and viral antigens. O.B. developed the sample preparation protocol. C.F. developed the sample preparation and assay protocol, supervised the lab work, and contributed to the experimental design. J.M.S. and N.R. conceived and supervised the study, and wrote the manuscript with input and feedback from all authors. The final version of the manuscript was approved by all co-authors.

## Funding

## Competing interests

OB and NR are co-founders of Capitainer AB, a company that commercialized the blood collection device for microsampling. All other authors declare no conflicts of interests.
