## [Peer Review File · Nature Communications]

Reviewers' Comments:

Reviewer #1:

Remarks to the Author:

The authors describe the development of a multiplexed serology assay for SARS-CoV-2 and its application in a Swedish study with home-sampled blood. This addresses two important issues, sampling outside clinical care and understanding the immune response in symptom-free/non-hospitalized individuals. The authors chose an unbiased, open approach to utilizing various SARS-2 protein preparations and several classification approaches. The link between severity of symptoms and number of detected SARS-CoV-2 proteins is interesting and merits further investigation.

The main challenge lies in the time span between generation and submission of this data. Much has been learned about SARS-2 serology in the meantime, and multiple routine clinical assays have been developed. It is understood that in this unusual situation of rapid scientific developments, manuscripts almost inevitably describe data that may seem a little outdated by the time of submission. However, the real value of the data presented in this manuscript is somewhat buried under the sheer breadth of analyses and the excessive supporting material which confuse the reader and distract from the main questions and findings, in my view: i) How can SARS-2 seroprevalence be estimated reliably using home-sampled blood (this is a major strength of the paper, as self-sampling of classical DBS is unreliable, as the authors correctly note). ii) What are the strengths of the multiplexed assay in comparison with N and S based assays that are routinely used nowadays (in early 2021, this is what readers will expect to see). iii) What can we learn from additional proteins? If the paper was slightly re-focused on its actual strengths, and shortened and streamlined to address these most important (and maybe other) questions, it would constitute a relevant and timely addition to the existing literature.

Specific recommendations include:

1. Please update the literature as best as possible.
2. Table 1 needs to be referenced early in the results. Provide accession numbers and amino acid ranges where possible. What are the differences between the protein preparations beyond their tags? If SPK_01 and SPK_02 (equally RBD_01 and RBD_02) are expression batches of the same protein, can they be treated as separate proteins in the UMAP and PCA analyses?
3. Discuss the generalizability of the results from the random population study for the city of Stockholm. The abstract seems to imply a seroprevalence of 12.5% whereas the discussion seems to consider the measured 5% with IgG against S the most relevant result.

Reviewer #2:

Remarks to the Author:

In their study "Multi-analyte profiling of home-sampled blood from randomly 1 selected individuals reveals diversity in the humoral immune 2 response against SARS-CoV-2" Roxhed et al. establish a protocol to use dried blood spots sampled at home to determine seroprevalences using a bead-based serological assay. Overall this is a timely manuscript that covers an important topic. Also, the data largely support the conclusions, there are some minor points that need to be revised before the manuscript can be considered for publication.

There are many graphical errors mainly in the supplementary data file. The file has been opened with Adobe Acrobat Reader which is the standard for any pdf. The authors should provide a file that contains the correct graphs.

Page 4 line 51-53: As far as I know most serological assays use the S protein derived antigens and only a subset uses N protein derived antigens.

The SCOV2 spike protein is commonly abbreviated as "S" and not SPK. This is confusing and should be changed throughout the manuscript and the supplementary figures

Supplementary figure S5: For validation of DBS samples vs Plasma samples against the same antigen, i.e. S1 and N, the authors must provide a correlation analysis and provide R2 values.

Figure S6 shows only empty graphs. This might be a graphical error and should be corrected

Figure S8 A and B shows also only empty graphs. Also there is no colour in C and D. Please correct
Page 6 line 102: What are the reasons for excluding 18% of the returned cards from analysis?
Figure 1D: I assume red dashed lines represent cut-offs? Or are these z-scores and the red dashed line represents the mean of the sample? Please indicate in the figure legend
Page 9 line 146: against which protein were the 6 negative control samples positive? Please specify. 6 false positive samples out of 95 is quite a lot. Did the authors exclude the rec. proteins used or corrected for this low(ish) specificity in the data analysis?
Could the authors please provide a demographic table also for the neg. ctrl. set?
Page 9 line 158: It not plausible why the authors did not find any antibodies for NL63 S1 since the seroprevalence in adults is usually above 90%. This indicates either a potential problem with the antigen or the cut-off used.
Page 9 line 162-163: What do the authors mean by 2 preparations of the SCOV2 proteins. Is it 2 expressions of recombinant proteins on different days or the same lot of rec proteins that are coupled to the beads at different days. Could the author please provide a coefficient of variation of repeat measurements between different protein productions and different bead couplings?
Discussion: The whole setup with analysing different antigen and different isotypes is very complicated and is therefore unlikely to be established in other labs. While I do not doubt the validity of the data, I would suggest that the authors discuss how they could simplify the analysis, i.e. which antigens/isotypes are the most important and which ones did not provide additional information.

Reviewer #3:

Remarks to the Author:

In "Multi-analyte profiling of home-sampled blood from randomly selected individuals reveals diversity in the humoral immune response against SARS-CoV-2" Roxhed et al. report on a cross-sectional observational study in 878 individuals in Stockholm, Sweden conducted in Spring 2020. From these individuals home-sampled dried blood spots were collected and profiled for IgG and IgM antibodies against several SARS-CoV-2 proteins. The authors report an accumulated seroprevalance of 12.5% and conclude that their approach is viable to assess the humoral immune response against the virus in the general population.

This manuscript outlines a wealth of information/data and is therefore (in part) hard to understand. On the other hand, the text is both poorly structured and. This is probably mainly due to the fact that it is not clear what the main objective/question of this study is. The authors constantly switch between aspects of biochemical basic research and the clinical application. Combined in one paper, both aspects are addressed in an unsatisfactory way. In sum and in my opinion, the paper does not meet the quality criteria necessary for a publication in NC. As a further indication of this note there is also a lack of care in the Supplement where some of the figures (e.g. S6) had no content or were damaged.

REVIEWER COMMENTS

Reviewer #1 (Remarks to the Author):

The authors describe the development of a multiplexed serology assay for SARS-CoV-2 and its application in a Swedish study with home-sampled blood. This addresses two important issues, sampling outside clinical care and understanding the immune response in symptom-free/non-hospitalized individuals. The authors chose an unbiased, open approach to utilizing various SARS-2 protein preparations and several classification approaches. The link between severity of symptoms and number of detected SARS-CoV-2 proteins is interesting and merits further investigation.

The main challenge lies in the time span between generation and submission of this data. Much has been learned about SARS-2 serology in the meantime, and multiple routine clinical assays have been developed. It is understood that in this unusual situation of rapid scientific developments, manuscripts almost inevitably describe data that may seem a little outdated by the time of submission. However, the real value of the data presented in this manuscript is somewhat buried under the sheer breadth of analyses and the excessive supporting material which confuse the reader and distract from the main questions and findings, in my view:

i) How can SARS-2 seroprevalence be estimated reliably using home-sampled blood (this is a major strength of the paper, as self-sampling of classical DBS is unreliable, as the authors correctly note).

ii) What are the strengths of the multiplexed assay in comparison with N and S based assays that are routinely used nowadays (in early 2021, this is what readers will expect to see).

iii) What can we learn from additional proteins? If the paper was slightly re-focused on its actual strengths, and shortened and streamlined to address these most important (and maybe other) questions, it would constitute a relevant and timely addition to the existing literature.

>> We thank the reviewer for these excellent comments and suggestions. We hope that the new version of our manuscript will meet the set expectations. .

Specific recommendations include:

1. Please update the literature as best as possible.

>> We thank the reviewer for raising this point, but giving the speed with which new serological data sets and studies are being published, we kindly ask to accept that we cannot find and include all work of relevance. We have added 5 additional publications from the last 6 months that use multiplexed serology and/or home sampling.

2. Table 1 needs to be referenced early in the results. Provide accession numbers and amino acid ranges where possible.

>> We have now updated Table 1A as requested and added accession numbers and related publications where available.

What are the differences between the protein preparations beyond their tags? If SPK_01 and SPK_02 (equally RBD_01 and RBD_02) are expression batches of the same protein, can they be treated as separate proteins in the UMAP and PCA analyses?

>> For the UMAP/PCA analysis, we chose to include all available proteins representing S, RBD and N antigens as well as combine data from measurements of IgG and IgM. Hence the proteins are being treated as independent features by the models. However, the data for paired antigen preparations were highly correlated ($\rho > 0.9$). The inclusion of several versions from one protein may indeed increase the weight over the ones where fewer versions were available. To illustrate the decomposed contribution from each antigen, we transposed the data input matrix for the PCA analysis for IgG and IgM detection, see below. While data from each isotype splits the dimensions, the paired antigen resides in proximity to another. This illustrates that they contribute with similar features to the data.

As pointed out in Table 1, the different preparations are both new batches of a protein from the same lab as well as proteins produced and provided by different

labs. The use of different preparations of the same protein was an initial effort to engage the community and account for the novelty of the target proteins in serological analysis. Using a threshold defined by two antigens revealing a positive or negative annotation (dual-antigen), we wanted to make use of the technology's multiplexing capacity and utilize several proteins to increase the stringency. Including several representatives indeed assisted us to increase the reliability of the observed findings.

3. Discuss the generalizability of the results from the random population study for the city of Stockholm. The abstract seems to imply a seroprevalence of 12.5% whereas the discussion seems to consider the measured 5% with IgG against S the most relevant result.

>> Thank you for pointing this out. We have now updated the Discussion section to provide some general implications of these observed numbers.

Reviewer #2 (Remarks to the Author):

In their study “Multi-analyte profiling of home-sampled blood from randomly 1 selected individuals reveals diversity in the humoral immune 2 response against SARS-CoV-2” Roxhed et al. establish a protocol to use dried blood spots sampled at home to determine seroprevalences using a bead-based serological assay. Overall this is a timely manuscript that covers an important topic. Also, the data largely support the conclusions, there are some minor points that need to be revised before the manuscript can be considered for publication.

>> We thank the reviewer for the very positive feedback and comments!

There are many graphical errors mainly in the supplementary data file. The file has been opened with Adobe Acrobat Reader which is the standard for any pdf. The authors should provide a file that contains the correct graphs.

>> We do apologize for the inconvenience and issues arising from the graphical representations. We had been notified by the editor and provided an updated version late into the first round of review. We do hope that all the figures and details are now visible as expected.

Page 4 line 51-53: As far as I know most serological assays use the S protein derived antigens and only a subset uses N protein derived antigens.

>> To our knowledge, the N protein has been far more often used in clinical assays (Roche, Abbott) during the first wave because the protein is easier to produce at scale. The N protein can be generated in bacterial expression systems, while the S proteins do require mammalian expression systems for glycosylation and folding to occur. It is only more recently that manufacturing the S proteins reached a scale that has allowed them to deploy these proteins for certified and test systems.

The SCOV2 spike protein is commonly abbreviated as “S” and not SPK. This is confusing and should be changed throughout the manuscript and the supplementary figures

>> We thank the reviewer for pointing this out, and we understand the reaction this may create at first. However, we have made a deliberate decision to name different batches of the same virus protein by using these acronyms. Our aim was not to confuse the reader with names too similar to the current terminology (such as S1.1 or S1.a) but to more clearly differentiate that each virus protein is represented by a different recombinant version. We would like to keep these descriptions as summarized in Table 1, the main text and figures. We remain open to the editor's decision to weigh in on this matter.

Supplementary figure S5: For validation of DBS samples vs Plasma samples against the same antigen, i.e. S1 and N, the authors must provide a correlation analysis and provide R2 values.

>> We thank the reviewer for pointing this out. We have now added two tables to the supplementary data file that reveal the correlation values between the DBS and plasma samples for all antigens. The correlations of each antigen was $R^2 > 0.9$ when comparing the data obtained in DBS and plasma (see sheet: “Paired sample pilot tests”) as well as when comparing paired antigens in the sample type (see

sheet: "Paired antigen pilot tests"). We consequently updated information in the supplementary file accordingly.

Figure S6 shows only empty graphs. This might be a graphical error and should be corrected

>> Thank you for pointing this out and please refer to the file with corrected figure representations.

Figure S8 A and B shows also only empty graphs. Also there is no colour in C and D. Please correct

>> Thank you for pointing this out and please refer to the file with corrected figure representations.

Page 6 line 102: What are the reasons for excluding 18% of the returned cards from analysis?

>> We only used DBS where the disc was sufficiently filled with blood samples. Samples excluded thereby represent the fraction of sampling cards where the study participant did not succeed in providing 10 μ l of blood. This has now been added and clarified in the method description.

"We examined each sampling card by visual inspection for the degree that the discs were filled with blood. Only the questionnaire data from donors providing at least one disc filled with 10 μ l of blood were then analyzed."

Figure 1D: I assume red dashed lines represent cut-offs? Or are these z-scores and the red dashed line represents the mean of the sample? Please indicate in the figure legend

>> A description has now been added. The red lines represent the centers of 0 (zero) of the relative antibody titers obtained from the normalized fluorescence data. In the Materials and Methods section it states:

"To minimize the influence of the less frequent seropositive samples, we used a robust regression model with a Huber loss function 31 consisting of an L2 loss for errors smaller than 0.1 and L1 loss for larger errors, as well as an L2 penalty on regression parameters. This procedure was implemented using the Julia package MLJLinearModels.jl and applied to normalize the data on the basis of the assay background obtained from the bare bead with the peak of the seronegative samples centered at zero. This normalization converted raw MFI data to residuals from this regression and denoted as relative antibody titers."

Page 9 line 146: against which protein were the 6 negative control samples positive? Please specify. 6 false positive samples out of 95 is quite a lot. Did the authors exclude the rec. proteins used or corrected for this low(ish) specificity in the data analysis?

>> We thank the reviewer for pointing out this important aspect. We revisited the data to present more insights about this matter to better explain our data. We agree that the way it was stated in the previous versions created the notion that the specificity was not high. To clarify, among the 95 negative controls, there were 6 individuals for which IgG levels were detected above the cut-off for any of the SARS-CoV-2 antigens, as presented below:

- 2 subjects were deemed IgG⁺ for dual-S = > 97.9% specificity
- 1 subject were deemed IgG⁺ for SPK_01 = > 98.9% specificity
- 2 subjects were deemed IgG⁺ for NCP_02 = > 97.9% specificity
- 1 subject was deemed IgG⁺ for NCP_01 = > 98.9% specificity

To avoid further confusion, we removed the sentences about these observations and added clarification by stating “For each antigen, the calculated sensitivity, specificity and seroprevalence levels can be found in the Supplementary Data File (sheet: “Prevalence per Antigen”). All examples had a borderline level and none of the negative were deemed IgG+ (or IgM+) by more than one protein. This clearly points at the utility of using different antigens for increasing the stringency to assess seroprevalence levels. Alternatively, an increased cut-off level can be applied.

Could the authors please provide a demographic table also for the neg. ctrl. set?

>> There were two sets of negative controls. There is no information about the 25 anonymous donors for which we purchased blood. Demographic information in terms of age, sex, and region of the 44 negative controls from patients have now been added to the manuscript.

Page 9 line158: It not plausible why the authors did not find any antibodies for NL63 S1 since the seroprevalence in adults is usually above 90%. This indicates either a potential problem with the antigen or the cut-off used.

>> We thank the reviewer for pointing this out. We have used a population-based model focusing on the assumption that only a subset of all participants would be seropositive for a new emerging virus. After processing the raw data to account for sample-specific differences in background levels (and thereby increasing the reliability of “true” signals above background), the majority of the relative antibody levels were centered around 0 (zero). In a population where most have already been exposed to a virus, our analysis approach will display commonly elevated antibody titers these at relative antibody levels around 0 (zero). This means that our approach retains the ranking within each analyte but it less suitable to compare levels across different analytes. For prevalent viruses, the approach can however be useful to identify the seronegative subjects. We foresee future efforts to include true positive and true negative controls for these viruses to demonstrate their applicability. At the time the presented study was conducted, there were no biobanked blood samples from individuals with confirmed infections from other viruses available for us. Hence, this would exceed the scope of the SARS-CoV-2 focused efforts.

However, as we have shown for the Epstein Barr virus (EBV), another prevalence virus circulating in the population, the data is centered at 0 (zero) and the relative antibody levels follow an inverted titer distribution compared to the SARS-CoV-2 antigens. Samples with deviating anti-EBNA antibody levels from the majority are found towards the lower levels as compared to SARS-CoV-2, where outliers were on the positive side of the scale.

To illustrate this, we show data from IgG levels against SARS-CoV-2 (SPK_02), EBV (EBN_01) and hCoV-NL63 (NLS_01). The first two rows of data are from unprocessed intensity levels (MFI) as provided by the Luminex output files, and presented in a linear scale (upper row) and after log₂-transformations (middle row). The bottom row is from background corrected processed data (Relative Antibody Levels), as found as Fig S9C. From these illustrations the differences between the distribution of antibody levels for new and

endemic viruses become explained. Clearly, a majority of samples contain antibodies against EBV and NL63, however, processing of the data account for this distribution and centers the majority data at 0 (zero).

As we have shown when comparing several antigen preparations, we cannot exclude the fact that the antigens performed optimal or as expected. We have added a necessary clarification to this matter into the results section “Antigen-centric assessment of seroprevalence in home-sampled blood.” and the discussion.

Page 9 line 162-163: What do the authors mean by 2 preparations of the SCOV2 proteins. Is it 2 expressions of recombinant proteins on different days or the same lot of rec proteins that are coupled to the beads at different days.

>> This refers to both, using proteins produced by two different labs as well as different batches of a protein produced by the same lab (see Table 1 for clarification). The beads may have been coupled on different days but combined into one bead array for the analysis and calling of seropositivity. We have added some more clarification to this matter.

Could the author please provide a coefficient of variation of repeat measurements between different protein productions and different bead couplings?

>> Please refer to Table S1 for the details about CVs for the different proteins in repeated measurements. Using the average CVs the values ranged from 10-15% and were very similar for the different preparations of the antigens. As exemplified for RBD_01 and RBD_02, we coupled new preparations from the same protein onto different beads and tested these together. As further listed in the Supplementary Data sheet "Paired antigens in pilot tests", there is a very high correlation between these proteins ($\rho = 0.99$).

Discussion: The whole setup with analysing different antigen and different isotypes is very complicated and is therefore unlikely to be established in other labs. While I do not doubt the validity of the data, I would suggest that the authors discuss how they could simplify the analysis, i.e. which antigens/isotypes are the most important and which ones did not provide additional information.

>> We thank the reviewer for this comment and it was indeed one task of our work to identify the most informative antigens among the S, RBD and N proteins. We hope the new version has now made it clearer that we as well as others (see updated list of references added to the manuscript) favor the S protein over N or RBD. It seemed to us that data from the S provide the most insightful information, however we see a great value in adding complementary information from other SARS-CoV-2 proteins as well. As we discussed, the use of dual-antigen scores for the S protein or combining the S and N proteins appeared most valuable.

Over the last few months, kits with multiple antigens have become commercially available to make use of these complementary information. While it indeed appears challenging to incorporate several features into the decision making, it actually resides within the data analysis models to process the information and reveal a score of interest (eg traffic light systems). We foresee that methods and systems will become available to detect different isotypes on top of the multiple antigens. It became clear that IgG is more useful to judge past infections but IgM can provide indications about a more recent exposure. This will ultimately provide the most insightful information to judge the state and phase of the immune response.

Reviewer #3 (Remarks to the Author):

In "Multi-analyte profiling of home-sampled blood from randomly selected individuals reveals diversity in the humoral immune response against SARS-CoV-2" Roxhed et al. report on a cross-sectional observational study in 878 individuals in Stockholm, Sweden conducted in Spring 2020. From these individuals home-sampled dried blood spots were collected and profiled for IgG and IgM antibodies against several SARS-CoV-2 proteins. The authors report an accumulated seroprevalance of 12.5% and conclude that their approach is viable to assess the humoral immune response against the virus in the general population.

>> We thank the reviewer for this correct and concise summary.

This manuscript outlines a wealth of information/data and is therefore (in part) hard to understand. On the other hand, the text is both poorly structured and. This is probably mainly due to the fact that it is not clear what the main objective/question of this study is.

>> We appreciate the positive perspective on our work, and acknowledge the challenge in compressing the information available from the large data set into the most interesting message. As our work combines aspects related several areas such as technology development, population epidemiology, Covid 19 serology and translational research, the main objective is the interplay between these rather than only one. In light with the comments from other reviewers and guides given by Reviewer #1, the revised version of our manuscript will now meet the request for clearer main objective.

The authors constantly switch between aspects of biochemical basic research and the clinical application.

>> We agree with the reviewer that these two main aspects may be confusing at times. However, the current pandemic has both required as well as given us new opportunities to connect basic research, technologies and clinical aspects in new a fashion. As requested by Reviewer #1, we have now revised our work and restructured the flow accordingly. We wish that these changes will better align our manuscript with the expectations from Reviewer #3.

Combined in one paper, both aspects are addressed in an unsatisfactory way. In sum and in my opinion, the paper does not meet the quality criteria necessary for a publication in NC.

>> We respectfully disagree with this assessment of our work and it remains unclear for us which quality criteria are being referred to and what the specific reasons for this assessment would be. Nonetheless, we do hope that the reviewer will be more convinced by the revised version of our manuscript.

As a further indication of this note there is also a lack of care in the Supplement where some of the figures (e.g. S6) had no content or were damaged.

>> We respectfully disagree that a formatting issue is reflecting quality. We excuse the issues with the supplementary figures, and upon the editors notification returned a revised pdf file on December 2nd to allow a complete review.

Reviewers' Comments:

Reviewer #1:

Remarks to the Author:

The authors have substantially streamlined and improved the manuscript, and this reviewer's suggestions were fully taken into account in the revision. I have no hesitation in recommending acceptance of the manuscript and its publication. The data presented therein make an important contribution to the literature dealing with SARS-2 seroepidemiology.

Reviewer #2:

Remarks to the Author:

The authors have addressed all comments of the reviewer sufficiently. In my opinion the manuscript can now be considered for publication.

Benjamin Meyer

Reviewer #3:

Remarks to the Author:

I thank the authors for the necessary and rather comprehensive revision of the manuscript. I have only one minor remark regarding the abstract: Your statement "link between severity...emerged"? on page 3, line 43 is uninformative and should either be skipped or be written/replaced with specific/appropriate statistics.

Reviewer #1 (Remarks to the Author): The authors have substantially streamlined and improved the manuscript, and this reviewer's suggestions were fully taken into account in the revision. I have no hesitation in recommending acceptance of the manuscript and its publication. The data presented therein make an important contribution to the literature dealing with SARS-2 seroepidemiology.

Reviewer #2 (Remarks to the Author): The authors have addressed all comments of the reviewer sufficiently. In my opinion the manuscript can now be considered for publication.

Benjamin Meyer

Reviewer #3 (Remarks to the Author): I thank the authors for the necessary and rather comprehensive revision of the manuscript. I have only one minor remark regarding the abstract: Your statement "link between severity...emerged"? on page 3, line 43 is uninformative and should either be skipped or be written/replaced with specific/appropriate statistics.

We thank the reviewer for acknowledging our efforts and we are very pleased to read that our revised version has addressed the constructive comments. The sentence in line 43 now refers to the statistical information provided in the results section and reads as follows:

"Subjects being IgG+ for several SARS-CoV-2 proteins reported influenza-like symptoms more frequently than those being IgG+ for only the S protein (OR=6.7; $p < 0.001$). Among all seropositive cases, 30% were asymptomatic."